# The association between the amino acid transporter LAT1, tumor immunometabolic and proliferative features and menopausal status in breast cancer

Gautham Ramshankar[1,2], Ryan Liu[2,3], Rachel J. Perry[2]*

**1** Irvington High School, Fremont, California, United States of America, **2** Departments of Cellular & Molecular Physiology and Internal Medicine (Endocrinology), Yale School of Medicine, New Haven, Connecticut, United States of America, **3** Cedar Park High School, Cedar Park, Texas, United States of America

* rachel.perry@yale.edu

**Data Availability Statement:** The data underlying the results presented in the study are available from the URLs included in the Methods, and

## Abstract

L-type Amino Acid Transporter 1 (LAT1) facilitates the uptake of specific essential amino acids, and due to this quality, it has been correlated to worse patient outcomes in various cancer types. However, the relationship between LAT1 and various clinical factors, including menopausal status, in mediating LAT1's prognostic effects remains incompletely understood. This is particularly true in the unique subset of tumors that are both obesity-associated and responsive to immunotherapy, including breast cancer. To close this gap, we employed 6 sets of transcriptomic data using the Kaplan-Meier model in the Xena Functional Genomics Explorer, demonstrating that higher LAT1 expression diminishes breast cancer patients' survival probability. Additionally, we analyzed 3′-Deoxy-3′-[18]F-Fluorothymidine positron emission tomography-computed tomography ([18]F-FLT PET-CT) images found on The Cancer Imaging Archive (TCIA). After separating all patients based on menopausal status, we correlated the measured [18]F-FLT uptake with various clinical parameters quantifying body composition, tumor proliferation, and immune cell infiltration. By analyzing a wealth of deidentified, open-access data, the current study investigates the impact of LAT1 expression on breast cancer prognosis, along with the menopausal status-dependent associations between tumor proliferation, immunometabolism, and systemic metabolism.

## Introduction

As the second leading cause of cancer deaths in women, breast cancer has become a major clinical and social burden, with annual out-of-pocket costs for breast cancer care in the U.S. exceeding $3 billion in 2019 [1]. Because breast cancer has high economic and social costs, it has become increasingly necessary to identify potential risk factors, biomarkers, and treatments. Nearly 30% of breast cancer cases are caused by modifiable risk factors like excess body weight and alcohol consumption [2]. Several of the modifiable risk factors that predispose to breast cancer converge on metabolism. Consequently, a key priority in the cancer field has

duplicated here: https://wiki.cancerimagingarchive.
net/pages/viewpage.action?pageId=30671268
https://xenabrowser.net/datapages/?cohort=TCGA
%20Breast%20Cancer%20(BRCA)&removeHub=
http%3A%2F%2F127.0.0.1%3A7222 https://
xenabrowser.net/datapages/?dataset=TCGA.BRCA.
sampleMap%2FBRCA_clinicalMatrix&host=https%
3A%2F%2Ftcga.xenahubs.net&removeHub=https
%3A%2F%2Fxena.treehouse.gi.ucsc.edu%3A443
https://xenabrowser.net/datapages/?dataset=
TCGA.BRCA.sampleMap%2FHiSeqV2&host=https
%3A%2F%2Ftcga.xenahubs.net&removeHub=http
%3A%2F%2F127.0.0.1%3A7222 https://
xenabrowser.net/datapages/?dataset=
TcgaTargetGtex_RSEM_Hugo_norm_count&host=
https%3A%2F%2Ftoil.xenahubs.net&removeHub=
https%3A%2F%2Fxena.treehouse.gi.ucsc.edu%
3A443 https://xenabrowser.net/datapages/?
dataset=desmedt2007_public%2Fdesmedt2007_
genomicMatrix&host=https%3A%2F%
2Fucscpublic.xenahubs.net&removeHub=https%
3A%2F%2Fxena.treehouse.gi.ucsc.edu%3A443
https://xenabrowser.net/datapages/?dataset=donor
%2Fexp_seq.all_projects.donor.USonly.xena.
tsv&host=https%3A%2F%2Ficgc.xenahubs.
net&removeHub=https%3A%2F%2Fxena.
treehouse.gi.ucsc.edu%3A443 https://
xenabrowser.net/datapages/?dataset=chin2006_
public%2Fchin2006Exp_genomicMatrix&host=
https%3A%2F%2Fucscpublic.xenahubs.
net&removeHub=https%3A%2F%2Fxena.
treehouse.gi.ucsc.edu%3A443 https://
xenabrowser.net/datapages/?dataset=miller2005_
public%2Fmiller2005_genomicMatrix&host=https
%3A%2F%2Fucscpublic.xenahubs.
net&removeHub=https%3A%2F%2Fxena.
treehouse.gi.ucsc.edu%3A443 All code can be
found at the URL below (also in the Methods):
https://github.com/gramshankar/
LAT1BreastCancer.

**Funding:** The authors are grateful for awards from
the Lion Heart Foundation and from the Yale
Cancer Center (both to R.J.P.), which supported
this research. The funders had no role in study
design, data collection and analysis, decision to
publish, or preparation of the manuscript.

**Competing interests:** The authors have declared
that no competing interests exist.

been to investigate tumor metabolism and how it can be affected by a patient's lifestyle. In the 1920s, Otto Warburg discovered that in order to sustain their energetic needs while prioritizing generating the biomass and nucleotides required for rapid proliferation and growth, cancer cells have greater metabolic demands than their benign counterparts. Because of this, oncogenic metabolism is characterized by heightened glycolytic metabolism, which necessitates greater uptake of glucose. This phenomenon is now called the Warburg Effect and has greatly shaped the field of tumor metabolism [3]. However, many years after Warburg's groundbreaking work identifying glucose metabolism as a key contributor to tumor pathogenesis, there remains relatively less investigation into the role of amino acid metabolism in tumor progression. The same can be said about amino acid metabolic reprogramming, the abnormal changes to amino acid uptake or metabolic pathways caused by tumor progression. However, past literature has shown that low concentrations of amino acids in the tumor microenvironment inhibit nearby immune cells, weakening immune responses to tumor cells and contributing to tumor progression [4, 5]. These data beg further investigation of the tumor- and/or immune cell-centric metabolic role of amino acids in the tumor microenvironment.

In order to leave the tumor interstitial compartment and undergo metabolism by tumor cells, amino acids must cross the plasma membrane with the help of amino acid transporters. Amino acid transporters can thus facilitate the uptake of amino acids to meet the metabolic needs of cancer cells, explaining why the expression of these transporters has been associated with the proliferation of cancer cells. One such transporter, L-type amino acid transporter 1 (LAT1) is particularly important in the amino acid transport process [4]. Encoded by the gene Solute Carrier Family 7 Member 5 (*SLC7A5*), LAT1 is a light-chain protein that heterodimerizes with its heavy-chain partner 4F2hc (*SLC3A2*) through a conserved disulfide bridge, forming the human LAT1-4F2hc complex. A sodium-independent transporter, LAT1 is an integral membrane protein that mediates the transport of large neutral amino acids like methionine, leucine, and histidine by exchanging them with intracellular glutamine [6]. LAT1 is unique in that it transports multiple essential amino acids, which cannot be synthesized by the human body and must be obtained through diet [7, 8]. Considering the dietary dependence of its transported molecules, LAT1 is a particularly intriguing target to participate in the links between lifestyle, systemic metabolism, and cancer.

Positron emission tomography-computed tomography (PET-CT) is a powerful tool in cancer metabolism research due to its ability to visualize thin slices of tissue in vivo and quantify cells' metabolic activity by measuring radiotracers like 3′-Deoxy-3′-[18]F-Fluorothymidine ([18]F-FLT). An analog of the nucleoside thymidine, [18]F-FLT is phosphorylated by the cytosolic enzyme thymidine kinase 1 (TK1) and taken up into the cell. During the S-phase of the cell cycle, TK1 is overexpressed nearly tenfold and [18]F-FLT uptake is at its highest. In this way, concentrations of [18]F-FLT and TK1 are elevated in cancer cells, making [18]F-FLT uptake a quantitative marker for tumor proliferation [9–12]. Ki-67 is a nuclear nonhistone protein, and because it is only expressed in cells that are not in the $G_0$ phase of the cell cycle, it can only be observed in actively-proliferating cells. This quality has made Ki-67 a classic proliferative marker for tumor cells [13], and is included in the datasets analyzed in the current report.

Past studies have demonstrated that menopausal status affects to what extent obesity is a risk factor for developing breast cancer. In multiple studies, obesity has been observed to have a protective relationship with breast cancer risk in premenopausal patients whereas it is a risk factor for breast cancer in postmenopausal patients [14]. Because of this, we segmented our analyses based on patients' menopausal statuses. We used body mass index (BMI) in kg/m$^2$ as a metric for obesity. By analyzing PET-CT scans of 58 patients from The National Cancer Imaging Archive (TCIA), we correlate patients' calculated [18]F-FLT uptake and Ki-67 index values to their BMIs to study the relationship between obesity and breast cancer [10, 15, 16].

To demonstrate the relationship between LAT1 and poorer health outcomes with a larger sample size, we leveraged RNA-seq data in the UCSC Xena Functional Genomics Explorer [17]. This allowed us to visualize the effect of LAT1 expression on breast cancer prognosis in premenopausal and postmenopausal patients. Ultimately, we used a similar workflow to our prior published work to examine the impact of *SLC7A5*, a gene with a drastically different role in metabolism, in breast cancer [18]. Our analyses reveal new insights into the associations between clinical variables (obesity, menopausal status), cell proliferation, infiltration with multiple immune cell subtypes, tumor LAT1 expression, and survival in breast cancer patients, which deepen our understanding of the bidirectional relationships that may inform interventional studies targeting these variables in individuals with breast cancer.

## Methods

### ¹⁸F-FLT PET-CT quantitative image analysis

Deidentified PET-CT images produced during the ACRIN 6688 clinical trial [10] were obtained from The Cancer Imaging Archive (TCIA). This dataset, "ACRIN-FLT-Breast (ACRIN 6688)", can be found here: https://wiki.cancerimagingarchive.net/pages/viewpage.action?pageId=30671268. Because only publicly available, deidentified data were analyzed, and participants, all of whom were adults, gave written consent for their data to be used, deidentified, in public repositories, separate ethical approval is not required for these or other datasets analyzed in this manuscript. Data sharing via TCIA is approved under the supervision of the University of Arkansas for Medical Sciences (UAMS) Institutional Review Board (IRB # 205568), and informed consent was provided by the patients for their data to be shared with TCIA; however, the details of the consent process are not available to us. Because only publicly available, deidentified data were analyzed, and we had no information about the patients whose data were analyzed, separate ethical approval was not sought. All data are submitted to the TGCA in accordance with the submitter's institutional policies, including IRB approval and informed consent provided by the patients for their data to be submitted, deidentified, to the TCGA. However, the details of the consent process are not available to us. We did not seek separate IRB approval because our use of these deidentified data are covered under these IRB approvals.

All data were accessed for research purposes between 2/3/23 and 7/1/2023. We analyzed the scans of all patients with a menopausal status, height, weight, and 5 clear CT slices (i.e., slices in which the primary breast tumor could be identified and its corresponding SUV values could be generated) present in the dataset. 58 of the 90 enrolled patients in the ACRIN clinical trial met these criteria, and all were analyzed. Of these 58 patients, 26 were premenopausal and 32 were postmenopausal. Scans taken at 3 different dates were available for most patients, and we used the earliest scan (from the baseline scanning which was defined to be 4 weeks before any treatment was administered) to minimize the chemotherapeutic effect of the treatment used in the clinical trial. Likewise, heights and weights measured on patients' first visits were used. These data were selected for analysis because breast cancer treatment often causes some weight gain [19–22], which may obscure differences in BMI that could promote proliferation.

The patients' images were uploaded to Fiji ImageJ and we used the PET-CT Viewer plugin to view and analyze them. After identifying the primary breast tumor on the PET image, we selected the tumor and used the Brown Fat Volume tool to draw fixed-volume spheres around the interior regions of interest (ROIs) on the CT slice. 5 slices were used from each patient's scan. SUV parameters were set at 2 to 15, and ¹⁸F-FLT uptake was calculated in the tumor tissue in the specified ROI. ¹⁸F-FLT uptake on PET-CT scans is measured by calculating and recording lean body mass-corrected standardized uptake values (SUV) of which there are 3

types: $SUV_{Mean}$, $SUV_{Max}$, and $SUV_{Peak}$. After positioning a fixed-volume sphere on a tumor, within the ROI, $SUV_{Mean}$ represents the average SUV, $SUV_{Max}$ indicates the maximum SUV, and $SUV_{Peak}$ corresponds to the SUV derived from a localized cluster of voxels with high uptake [10, 23]. The primary endpoint of image analysis was BMI (kg/m$^2$) correlated to the 3 types of tumor SUV (g/mL).

## LAT1 prognostic analysis

Using the UCSC Xena Functional Genomics Browser (https://xenabrowser.net/), we accessed the "TCGA Breast Cancer (BRCA) cohort" (found here: https://xenabrowser.net/datapages/?cohort=TCGA%20Breast%20Cancer%20(BRCA)&removeHub=http%3A%2F%2F127.0.0.1%3A7222) which included 2 datasets. The BRCA cohort had 1247 total patients, and all of them had menopausal statuses recorded, which we accessed through the "Phenotypes" dataset: https://xenabrowser.net/datapages/?dataset=TCGA.BRCA.sampleMap%2FBRCA_clinicalMatrix&host=https%3A%2F%2Ftcga.xenahubs.net&removeHub=https%3A%2F%2Fxena.treehouse.gi.ucsc.edu%3A443. 1236 of the 1247 patients had survival data recorded. The "IlluminaHiSeq" dataset was used to study LAT1 expression and it can be found here: https://xenabrowser.net/datapages/?dataset=TCGA.BRCA.sampleMap%2FHiSeqV2&host=https%3A%2F%2Ftcga.xenahubs.net&removeHub=http%3A%2F%2F127.0.0.1%3A7222. The "IlluminaHiSeq" dataset used fragments per kilobase of exon per million mapped fragments (FPKM) to measure gene expression. 1218 of the 1247 patients had LAT1 expression data. These datasets were used alongside the Kaplan-Meier model in the Xena visualization suite to analyze LAT1 and its effect on breast cancer prognosis.

To analyze LAT1 expression, the following workflow was used: the 1247 patients were added to Column A. *SLC7A5* was added to Column B as a genomic variable with the gene expression dataset selected, and menopause status was added to Column C as a phenotypic variable. After removing null and duplicate samples, a Kaplan-Meier (KM) plot was generated in Column B to show LAT1 expression and its effect on prognosis in 1005 of the 1247 patients. Next, low and high-expression groups were created from these patients. After 34 patients with indeterminate menopausal statuses were removed from the dataset, 10.46 FPKM was calculated by Xena to be the median for LAT1 expression. Patients were divided at the median: 485 patients were in the low expression group ($< 10.46$ FPKM), and 486 patients were in the high expression group ($> = 10.46$ FPKM). A KM plot was generated for each group using Column C, creating 2 KM plots with the premenopausal, perimenopausal, and postmenopausal patients in each expression group.

In addition, after selecting a menopausal status (premenopausal or postmenopausal patients), a breast cancer subtype (ER+ or HER2-) was chosen, and survival was observed in low and high LAT1 expression groups. Data from patients with ER+ and HER2- tumors were also analyzed and a KM plot was created for each subtype's high and low expression groups, separately for pre- and postmenopausal patients. Patients were divided at the calculated median expression level for each group: 10.09 FPKM for the ER+ patients and 10.38 FPKM for the HER2- patients.

In addition to the BRCA dataset, the following datasets were used to access breast cancer patients' gene expression data: "RSEM norm-count" from the "TCGA TARGET GTEx" cohort (https://xenabrowser.net/datapages/?dataset=TcgaTargetGtex_RSEM_Hugo_norm_count&host=https%3A%2F%2Ftoil.xenahubs.net&removeHub=https%3A%2F%2Fxena.treehouse.gi.ucsc.edu%3A443), "Desmedt 76 Gene Node-Neg Gene Exp" from the "node-negative breast cancer (Desmedt 2007)" cohort (https://xenabrowser.net/datapages/?dataset=desmedt2007_public%2Fdesmedt2007_genomicMatrix&host=https%3A%2F%2Fucscpublic.

xenahubs.net&removeHub=https%3A%2F%2Fxena.treehouse.gi.ucsc.edu%3A443), "gene expression RNAseq—US projects" from the "ICGC (donor centric)" cohort (https://xenabrowser.net/datapages/?dataset=donor%2Fexp_seq.all_projects.donor.USonly.xena.tsv&host=https%3A%2F%2Ficgc.xenahubs.net&removeHub=https%3A%2F%2Fxena.treehouse.gi.ucsc.edu%3A443), "Gene Expression" from the "Breast Cancer (Chin 2006)" cohort (https://xenabrowser.net/datapages/?dataset=chin2006_public%2Fchin2006Exp_genomicMatrix&host=https%3A%2F%2Fucscpublic.xenahubs.net&removeHub=https%3A%2F%2Fxena.treehouse.gi.ucsc.edu%3A443), and "Miller TP53 Gene Exp" from the "Breast Cancer (Miller 2005)" cohort (https://xenabrowser.net/datapages/?dataset=miller2005_public%2Fmiller2005_genomicMatrix&host=https%3A%2F%2Fucscpublic.xenahubs.net&removeHub=https%3A%2F%2Fxena.treehouse.gi.ucsc.edu%3A443). To measure expression, the "TCGA TARGET GTEx" dataset used FPKM, the "ICGC (donor centric)" dataset used normalized read count, and the "Node-negative breast cancer (Desmedt 2007)" and "Breast Cancer (Miller 2005)" datasets used log2 units. Although these gene expression datasets did not include menopausal status as a possible phenotypic variable, our selection criteria were to include breast cancer datasets that had RNA-seq data on SLC7A5 and survival data from the same patients that could be used to produce Kaplan-Meier plots on the Xena platform. Each dataset was used in the same way: SLC7A5 was selected as a genomic variable in Column B, and after null and duplicate samples were removed, a KM plot was generated. For the "TCGA TARGET GTEx" and "ICGC (donor centric)" datasets, only patients with breast tumors were selected for the analysis.

All of the KM plots were created with Overall Survival as the dependent variable unless otherwise specified. After patients were split at the median for the gene expression analyses, some groups had an unequal number of patients because patients with the same expression levels were put in the same group. Some patients were at the expression median, and the median was calculated to ensure they were placed in the high-expression group while keeping the sizes of each expression group roughly the same.

## Statistical analysis

Correlation tests were performed between patients' SUV and BMI values. 26 premenopausal patients' BMIs ranged from 23.829 to 142.822 kg/m$^2$ (mean [SD] = 33.972 [22.584]), and 32 postmenopausal patients' BMIs ranged from 17.940 to 199.219 kg/m$^2$ (mean [SD] = 40.865 [36.818]). The unusually high BMI values are driven by unusually low heights reported for these patients; however, only one premenopausal and postmenopausal patient had a reported BMI above 100. Because 5 slices were used per patient, each patient had 5 SUV$_{Max}$ values and 5 SUV$_{Peak}$ values but only 1 BMI. In order to correlate BMI and SUV, we needed the same number of values for each. In order to get one SUV for each patient, we took the mean of the SUVs produced from all 5 slices. For SUV$_{Mean}$, the calculated SUV had a margin of error indicated by a plus-minus sign. This meant that the calculation of each SUV$_{Mean}$ yielded 2 numerical values, one being the high value and the other being the low value, so 5 slices yield 10 SUV$_{Mean}$ values per patient. We took the mean of these 10 values for each patient. Each of these individual SUVs was then correlated with each patient's BMI.

We also correlated each patient's 3 types of SUVs to their Ki-67 values to further inform the validity of [18]F-FLT uptake as a metric for tumor proliferation. BMI was also correlated to Ki-67. All correlations were two-tailed Pearson correlation tests performed after patients' data were segmented by menopausal status. Shapiro-Wilk tests were also performed to determine if any groups of data were normally distributed. Student's t-tests and Mann-Whitney U tests

were performed on parametric and nonparametric data, respectively, to assess difference. For both tests, all data were transformed using log2 fold changes of the mean.

Unless otherwise specified, statistical analysis was done and graphs were made in Python 3.9 using the pandas (version 1.5) and SciPy (version 1.10) libraries. The two-tailed Pearson correlation tests were conducted using the "pearsonr" function from the scipy.stats module. The Mann-Whitney U tests were conducted using the "mannwhitney" function, the Student's t-tests were conducted using the "ttest_ind" function, and the Shapiro-Wilk tests were performed using the "shapiro" function, all from the scipy.stats module. All Python code can be found here: https://github.com/gramshankar/LAT1BreastCancer. For each of the KM plots, a log-rank test was conducted by Xena to compare the curves in the graph. Test statistics and p-values were calculated. Statistical significance was indicated by p-values less than 0.05, and marginally significant results have p-values greater than 0.05 but less than 0.10.

## Results

### Correlation analysis between proliferative markers, obesity, and immune cells by menopausal status

In premenopausal patients, Ki-67 insignificantly positively correlated with $SUV_{Mean}$, $SUV_{Peak}$ (marginal significance), and $SUV_{Max}$ (marginal significance) (Fig 1A). However, the relationship between Ki-67 and tumor $^{18}$F-FLT uptake was statistically stronger in postmenopausal patients, in whom Ki-67 significantly positively correlated with $SUV_{Mean}$, $SUV_{Peak}$, and $SUV_{Max}$ (Fig 1B).

In premenopausal patients, BMI insignificantly negatively correlated with $SUV_{Mean}$, $SUV_{Peak}$, and $SUV_{Max}$ (Fig 1A). In postmenopausal patients, BMI insignificantly positively correlated with $SUV_{Mean}$ (marginal significance), $SUV_{Peak}$, and $SUV_{Max}$ (Fig 1B). In

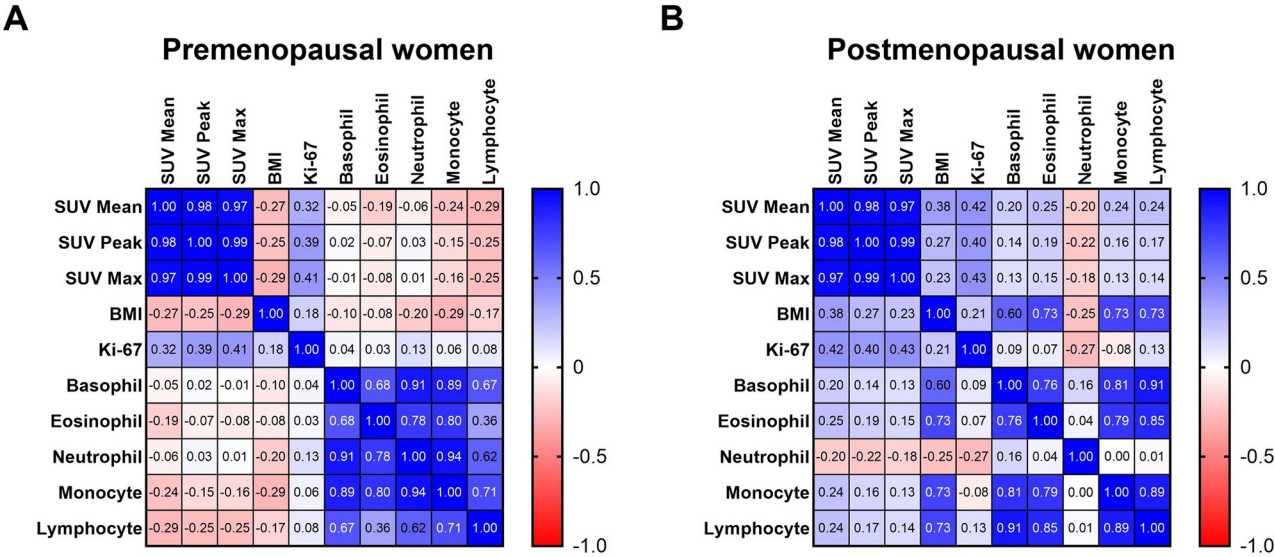

**Fig 1. Correlations between clinical variables.** Proliferative markers (Ki–67 and lean body mass–corrected $^{18}$F–FLT uptake measured by $SUV_{Mean}$, $SUV_{Peak}$, and $SUV_{Max}$), obesity (BMI), and immune cell counts (basophil, eosinophil, neutrophil, monocyte, and lymphocyte) were correlated in **(A)** premenopausal and **(B)** postmenopausal patients. Pearson r values were calculated and correlation matrices were generated in GraphPad Prism version 9.5.1.

premenopausal and postmenopausal patients, BMI insignificantly positively correlated with Ki-67 (Fig 1A and 1B).

In premenopausal patients, basophil, eosinophil, neutrophil, monocyte, and lymphocyte counts insignificantly negatively correlated with $SUV_{Mean}$, $SUV_{Peak}$, and $SUV_{Max}$. All immune cells insignificantly negatively correlated with Ki-67 (Fig 1A).

In postmenopausal patients, basophil, eosinophil, monocyte, and lymphocyte counts insignificantly positively correlated with $SUV_{Mean}$, $SUV_{Peak}$, and $SUV_{Max}$. Neutrophil counts insignificantly negatively correlated with $SUV_{Mean}$, $SUV_{Peak}$, and $SUV_{Max}$. Basophil, eosinophil, monocyte, and lymphocyte counts insignificantly negatively correlated with Ki-67. Neutrophil counts insignificantly positively correlated with Ki-67 (Fig 1B).

In premenopausal patients, BMI negatively correlated with basophil, eosinophil, monocyte, and lymphocyte counts and positively correlated with neutrophil counts (Fig 1A). Opposite relationships were observed in postmenopausal patients, in whom BMI positively correlated with basophil, eosinophil, monocyte, and lymphocyte counts and negatively correlated with neutrophil counts (Fig 1B).

These correlation tests' p-values are presented in Tables 1–4.

**Table 1. P–values from correlations between Ki–67 and $^{18}$F–FLT uptake.** Significant results are bolded and marginally significant results are bolded and italicized.

|  | Premenopausal Patients | Postmenopausal Patients |
|---|---|---|
| $SUV_{Mean}$ | 0.114 | **0.035** |
| $SUV_{Peak}$ | *0.081* | **0.045** |
| $SUV_{Max}$ | *0.078* | **0.029** |

**Table 2. P–values from correlations between BMI and proliferative markers.** Marginally significant results are bolded and italicized.

|  | Premenopausal Patients | Postmenopausal Patients |
|---|---|---|
| $SUV_{Mean}$ | 0.279 | *0.095* |
| $SUV_{Peak}$ | 0.335 | 0.342 |
| $SUV_{Max}$ | 0.268 | 0.437 |
| Ki-67 | 0.767 | 0.691 |

**Table 3. P–values from correlations between immune cells and proliferative markers in premenopausal patients.**

|  | Basophils | Eosinophils | Neutrophils | Monocytes | Lymphocyte |
|---|---|---|---|---|---|
| $SUV_{Mean}$ | 0.395 | 0.539 | 0.334 | 0.190 | 0.472 |
| $SUV_{Peak}$ | 0.401 | 0.620 | 0.420 | 0.262 | 0.404 |
| $SUV_{Max}$ | 0.519 | 0.691 | 0.520 | 0.342 | 0.537 |
| Ki-67 | 0.394 | 0.712 | 0.626 | 0.417 | 0.132 |

**Table 4. P–values from correlations between immune cells and proliferative markers in postmenopausal patients.**

|  | Basophils | Eosinophils | Neutrophils | Monocytes | Lymphocyte |
|---|---|---|---|---|---|
| $SUV_{Mean}$ | 0.239 | 0.237 | 0.671 | 0.235 | 0.233 |
| $SUV_{Peak}$ | 0.560 | 0.563 | 0.586 | 0.560 | 0.561 |
| $SUV_{Max}$ | 0.682 | 0.685 | 0.741 | 0.679 | 0.681 |
| Ki-67 | 0.781 | 0.737 | 0.986 | 0.723 | 0.762 |

## LAT1 expression and survival probability

In the TCGA BRCA gene expression dataset, patients in the high LAT1 expression group experienced lower overall survival than patients in the low expression group until 4000 days after initial treatment. From that point until 6500 days and again from 6600 days until 7500 days, the low LAT1 expression group had a worse overall survival rate (Fig 2A). Similarly, patients with high LAT1 expression had a lower disease-specific survival rate than patients with low LAT1 expression until 4400 days; after that point until the end of the study, patients in the low expression group had a lower disease-specific survival rate (Fig 2B). We recognize that the long follow-up period prevents us from concluding with certainty that mortality is breast cancer-related, but even if mortality were unrelated to cancer at this time point, the utility of LAT1 as a prognostic factor remains important.

In the "TCGA TARGET GTEx" gene expression dataset, patients in the high LAT1 expression group ($> = 10.65$ FPKM) experienced lower survival rates than the low LAT1 expression group ($< 10.65$ FPKM) until 4000 days after initial treatment. From 4000 days until the end of the study, the low-expression group had a lower survival rate (Fig 3A). In the "Node-negative breast cancer (Desmedt 2007)" gene expression dataset, after the first 500 days, the high LAT1 expression group ($> = 0.1221$ log2) experienced a lower survival rate than the low LAT1 expression group ($< 0.1221$ log2) for the remainder of the study (Fig 3B). In the "ICGC (donor centric)" gene expression dataset, the high LAT1 expression group ($> = 0.00002400$) had lower survival than the low LAT1 expression group ($< 0.00002400$) for the entire study. The high expression group's survival probability reached 0% near 4000 days (Fig 3C). In the "Breast Cancer (Chin 2006)" gene expression dataset, except from 1.2 to 1.4 years, the high LAT1 expression group experienced a lower survival rate than the low LAT1 expression group. Units were not given for this study but it most likely used log2 units (Fig 3D). In the "Breast Cancer (Miller 2005)" gene expression dataset, overall survival data were not available so

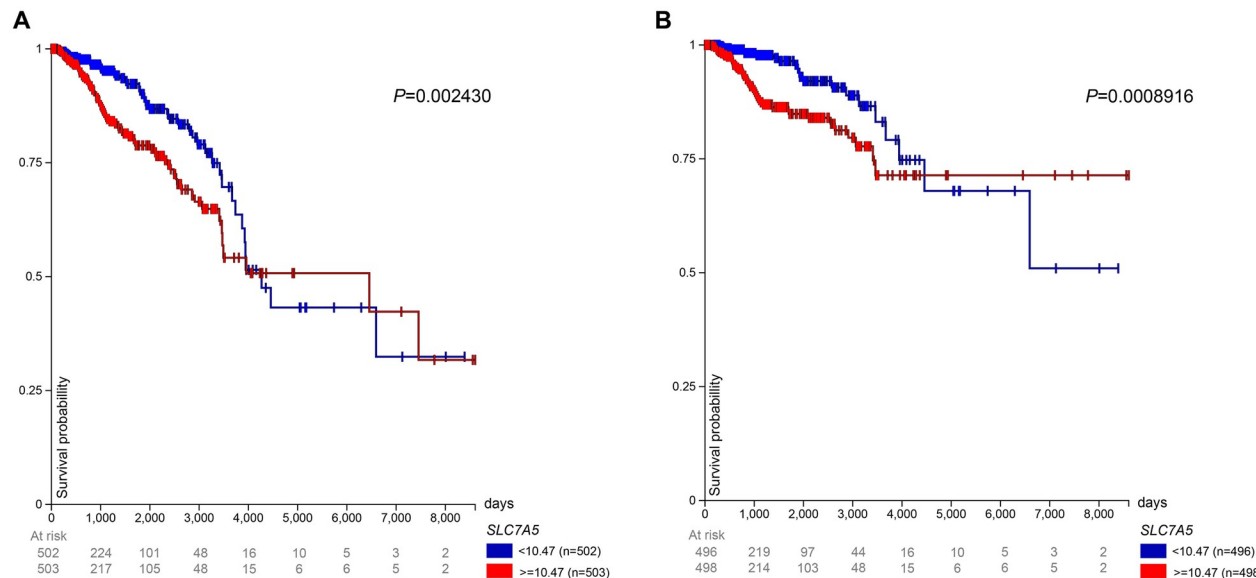

**Fig 2. LAT1 expression and prognosis in TCGA BRCA patients.** Prognosis in patients from the TCGA BRCA gene expression dataset. Patients were separated into high ($> = 10.47$ FPKM) and low ($< 10.47$ FPKM) LAT1 expression groups, and **(A)** overall survival and **(B)** disease–specific survival were observed up to 8605 days after initial treatment. The median for expression level is slightly different from the median stated earlier because patients with indeterminate menopausal status were included in this analysis.

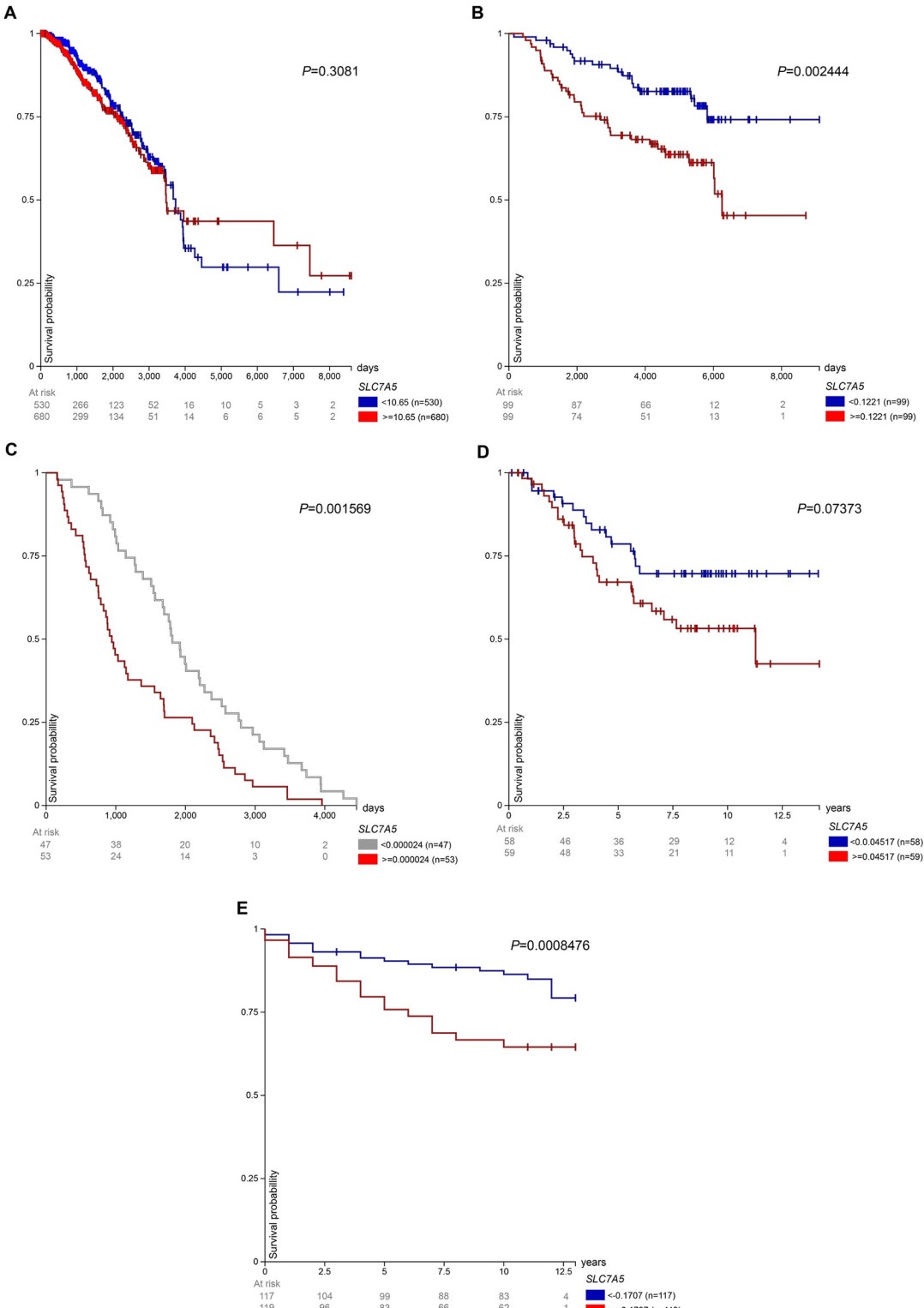

**Fig 3. LAT1 expression and prognosis in patients from other datasets.** After patients were separated into high and low LAT1 expression groups, survival was observed in patients from the **(A)** "TCGA TARGET GTEx", **(B)** "Node–negative breast cancer (Desmedt 2007)", **(C)** "ICGC (donor centric)", **(D)** "Breast Cancer (Chin 2006)", and **(E)** "Breast Cancer (Miller 2005)" cohorts.

disease-specific survival was observed. The high LAT1 expression group ($>$ = -0.1707 log2) experienced worse survival than the low LAT1 expression group ($<$ -0.1707 log2) for the entire study (Fig 3E). Overall, 2 of the gene expression datasets showed that low LAT1 expression conferred a poorer prognosis in breast cancer patients than high LAT expression, while 4 others showed the opposite.

## Survival probability with high LAT1 expression TCGA BRCA patients by menopausal status

The impact of menopausal status on survival in patients with high LAT1 expression is shown in Fig 4. After 1000 days, postmenopausal patients had the lowest survival rates. Premenopausal patients had the highest survival rates among the 3 groups until approximately 2500 days, from which point peri-menopausal patients had the highest survival until 3600 days (Fig 4); however, our ability to draw conclusions regarding survival in peri-menopausal patients is limited by the relatively low number of patients in this group.

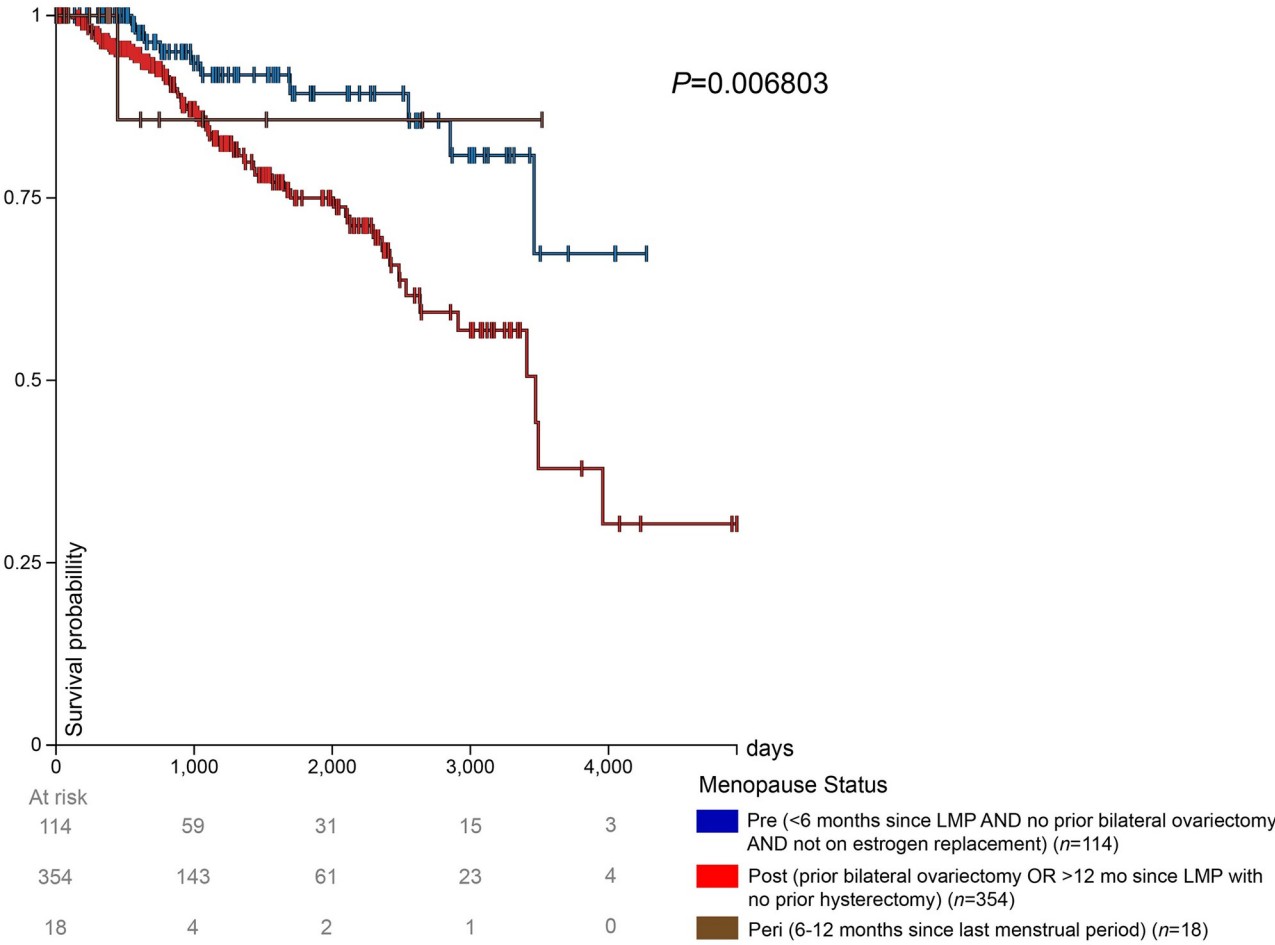

**Fig 4. High expression and prognosis in TCGA BRCA patients.** Prognosis in the high expression group ($>$ = 10.46 FPKM) from the TCGA BRCA gene expression dataset. The high expression group was separated into 3 groups: premenopausal, postmenopausal, and peri–menopausal breast cancer patients.

## Survival probability with low LAT1 expression TCGA BRCA patients by menopausal status

The impact of menopausal status on survival in patients with low LAT1 expression is shown in Fig 5. Premenopausal patients had a higher survival rate than postmenopausal patients until 3200 days. Postmenopausal patients, after 3200 days and until the end of the available survival data for premenopausal patients at approximately 3800 days, exhibited a higher survival rate compared to premenopausal patients. The few peri-menopausal patients in this study maintained a 100% survival probability throughout the duration that they were monitored.

## Survival probability of premenopausal vs. postmenopausal patients with ER+ tumors by LAT1 expression level

Fig 6 shows the survival of ER+ patients by menopausal status and LAT1 expression level. Among the premenopausal patients, the low expression group ($< 10.26$ FPKM) experienced a lower survival rate than the high expression group ($> = 10.26$ FPKM), excluding a short interval from approximately 2500 to 2900 days after initial treatment. Among the postmenopausal patients, the high expression group ($> = 10.07$ FPKM) experienced lower survival until

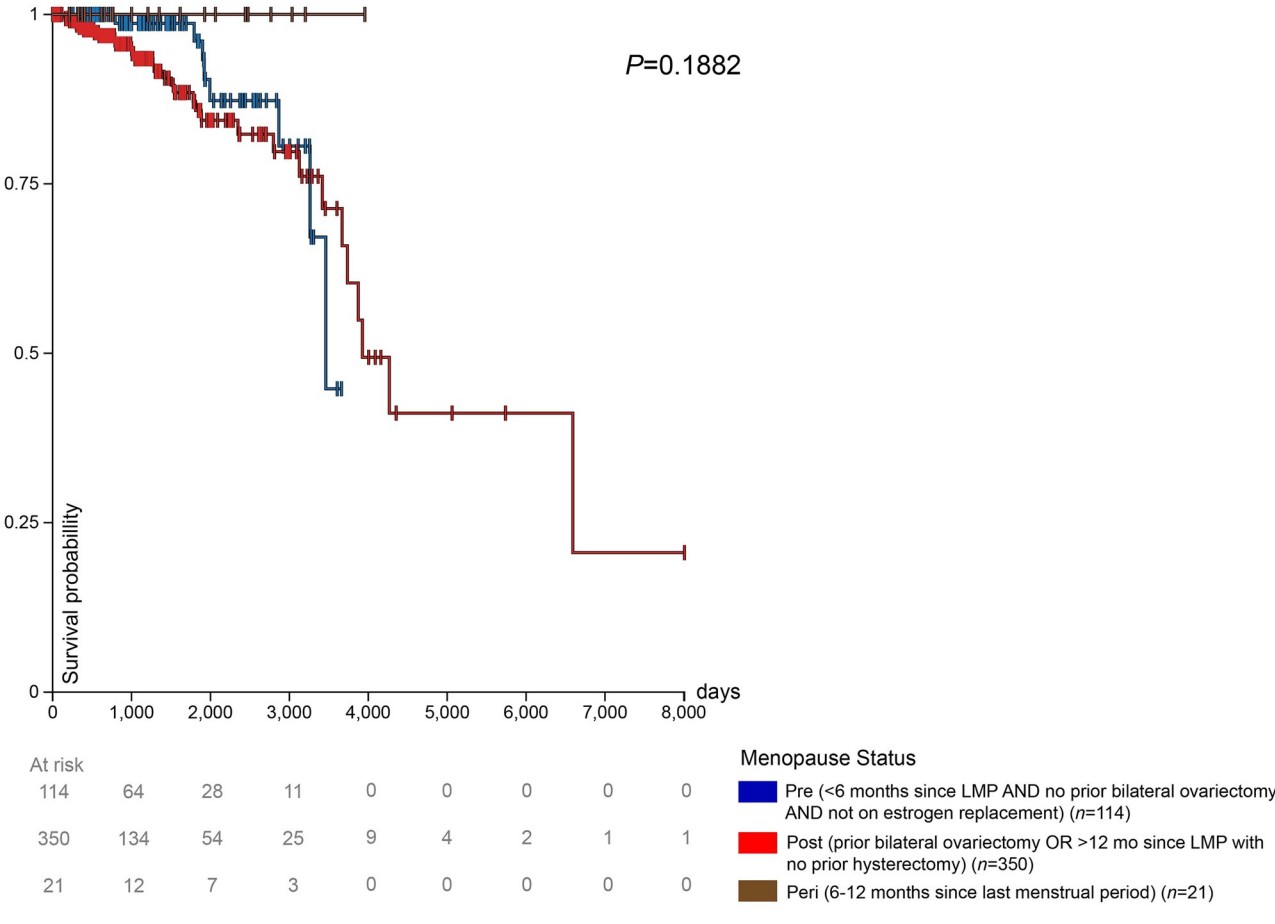

**Fig 5. Low expression and prognosis in TCGA BRCA patients.** Prognosis in the low expression group ($< 10.46$ FPKM) from the TCGA BRCA gene expression dataset. The low expression group was separated into 3 groups: premenopausal, postmenopausal, and peri–menopausal breast cancer patients.

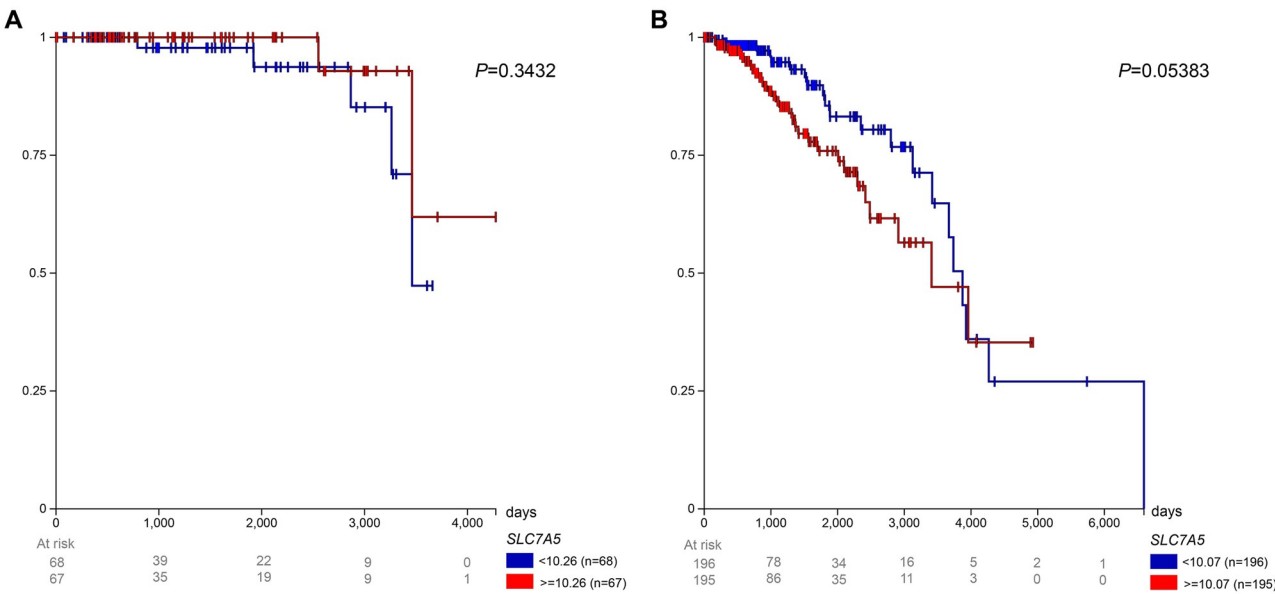

**Fig 6. Prognosis in TCGA BRCA patients with ER+ tumors.** Survival for (A) premenopausal and (B) postmenopausal patients was observed with low (< 10.26 FPKM for premenopausal and < 10.07 FPKM for postmenopausal) and high (> = 10.26 FPKM for premenopausal and > = 10.07 FPKM for postmenopausal) LAT1 expression.

approximately 3900 days after initial treatment, from which point the low expression group (< 10.07 FPKM) has a lower survival rate. The low expression group reached a 0% survival rate 4275 days after initial treatment.

## Survival probability of premenopausal vs. postmenopausal patients with HER2- tumors by LAT1 expression level

Fig 7 shows the survival of breast cancer patients with HER2- tumors by menopausal status and LAT1 expression level. Among premenopausal patients, the high expression group (> = 10.43 FPKM) experienced worse survival until nearly 3300 days after initial treatment. After then, the low expression group (< 10.43 FPKM) experienced worse survival until the end of the study. Among postmenopausal patients, the high expression group (> = 10.37 FPKM) experienced lower survival until approximately 3900 days after initial treatment, from which point the low expression group (< 10.37 FPKM) had worse survival until the end of the survival.

## Survival probability with low and high LAT1 expression in ER+ and HER2- tumors of TCGA BRCA patients by menopausal status

Next, we reanalyzed the same data shown in Figs 6 and 7 with different groupings, in order to demonstrate the impact of menopausal status on survival probability in patients with tumors expressing LAT1 at low (<10.09 FPKM) and high (> = 10.09 FPKM) levels. In patients with ER+ tumors with low LAT1 expression, postmenopausal patients experienced lower survival than their premenopausal counterparts, excluding a short interval from approximately 3200 to 3800 days after initial treatment (S1A Fig). The survival rate for postmenopausal patients reached 0% 6593 days after the start of the study. In the group with ER+ tumors and high LAT1 expression, postmenopausal patients experienced a considerably lower survival rate

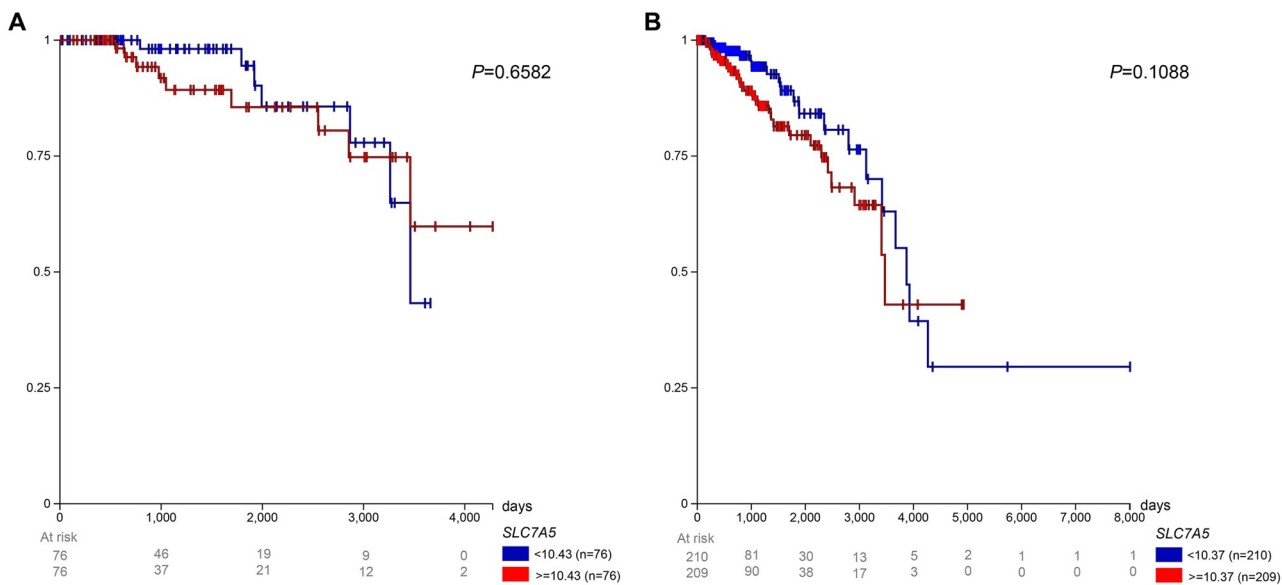

**Fig 7. Prognosis in TCGA BRCA patients with HER2– tumors.** Survival for (A) premenopausal and (B) postmenopausal patients was observed with low (< 10.43 FPKM for premenopausal and < 10.37 FPKM for postmenopausal) and high (> = 10.43 FPKM for premenopausal and > = 10.37 FPKM for postmenopausal) LAT1 expression.

than premenopausal patients (S1B Fig). In both expression groups, the small number of peri-menopausal patients with data available had a 100% survival rate throughout the study.

S2 Fig shows the survival of patients with HER2- breast tumors by menopausal status and tumor LAT1 expression level. Among patients with low LAT1 expression (< 10.38 FPKM), postmenopausal patients with HER2- tumors had a lower survival rate than premenopausal patients, excluding a short interval from approximately 3300 to 3700 days after initial treatment. Among patients with high LAT1 expression (> = 10.38 FPKM), postmenopausal patients with HER2- tumors experienced a lower survival throughout the study. In both expression groups, peri-menopausal patients had a 100% survival rate.

## Discussion

Increasing interest in the relationship between systemic metabolism, tumor metabolism, immunometabolism, and cancer outcomes, alongside evolving technologies expanding both the available data and the community's ability to mine it to develop new insights. To that end, in this study, we utilized multiple publicly available breast cancer datasets, including "ACRIN-FLT-Breast (ACRIN 6688)", TCGA BRCA "Phenotypes", TCGA BRCA "IlluminaHiSeq", "TCGA TARGET GTEx", "Node-negative breast cancer (Desmedt 2007)", "ICGC (donor centric)", "Breast Cancer (Chin 2006)", and "Breast Cancer (Miller 2005)", aiming to better understand the intersection between parameters of systemic metabolic health, tumor gene expression, and immune cell infiltration, and outcomes in individuals with breast cancer (Fig 8).

As opposed to genes or metabolic fluxes involved in glucose [24–31] or lipid metabolism [31–39], there exists a relative paucity of studies exploring the impact of expression of genes regulating amino acid uptake in breast cancer. Therefore, we elected to focus the current study on the expression of LAT1, which transports large amino acids including leucine, isoleucine, valine, phenylalanine, methionine, tyrosine, histidine, and tryptophan into the cell, and its relationships with body weight, tumor cell proliferation, and immune infiltration. Prior

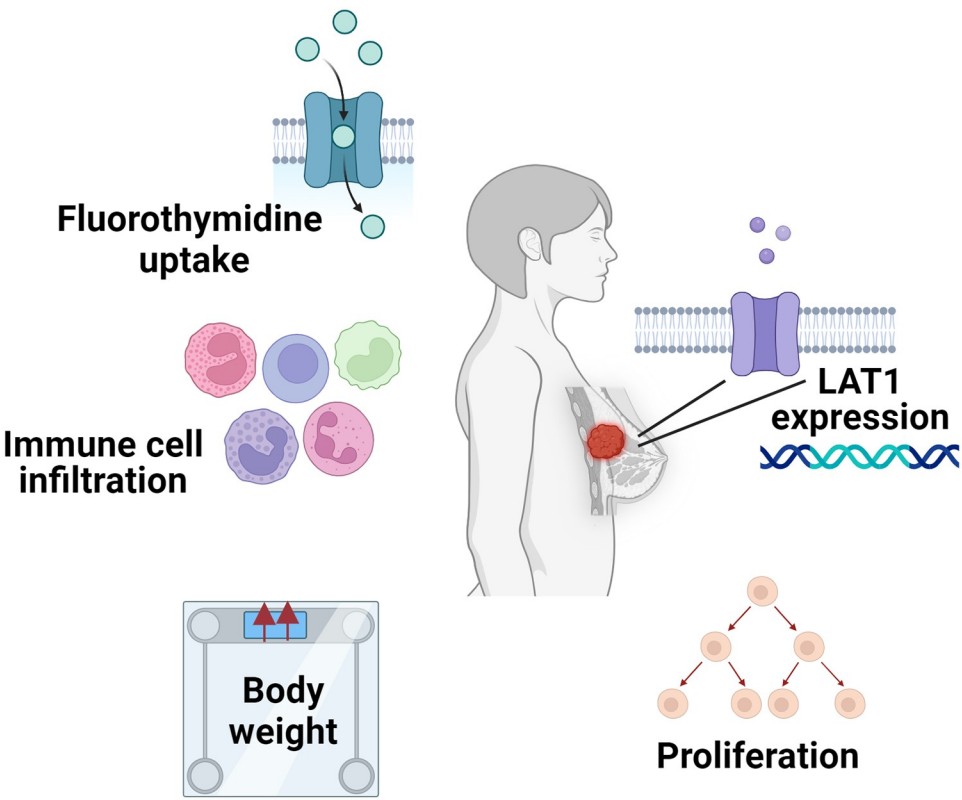

**Fig 8. Summary of factors correlated in this analysis.** Figure created with BioRender.com. The current study analyzed [18]F–FLT uptake, immune infiltrate levels, body weights, LAT1 expression, and tumor proliferative factors observed in breast cancer patients.

literature indicates that LAT1 is involved in protein synthesis [40, 41] and mTORC1 activity [42, 43], and may also modulate the anti-tumor immune response [44–47]. Overexpression of LAT1 has been observed in a plethora of tumor types ranging from lung to endometrial to liver, but fewer studies of the relationship between LAT1 and breast cancer exist [48]. Furthermore, LAT1 has been less frequently associated with a poor long-term clinical prognosis in breast cancer than in other cancers. Our data, too, provide mixed evidence: while some datasets showed that high LAT1 expression was worse for prognosis, others showed the opposite. Namely, the BRCA dataset showed that low LAT1 expression conferred worse survival at some points. This is likely because when compared to the high-expression group, a greater proportion of low-expression patients in the BRCA dataset had a positive margin status. Also, a greater percentage of the low-expression group had a distant metastasis present. The fact that the low LAT1 expression group tended to have more positive margin status and distant metastases than patients in the high-expression group may contribute to the discrepancy between our data on the predictive value of LAT1, as these have shown to be poor prognostic factors in breast cancer [49, 50]. Additionally, in the "TCGA TARGET GTEx" dataset, the low-expression group also had lower survival than the high-expression group. Margin status and the presence of distant metastases could also be confounding variables in this dataset, but data were not available to determine that. In this way, we show that the relationship between LAT1 expression and survival in breast cancer patients may be more complicated than previously appreciated.

Past analyses on LAT1 are not stratified by menopausal status, another unique quality of our study. In breast cancer patients with high tumor LAT1 expression, we observed worse survival in postmenopausal individuals as compared to peri- or premenopausal, but interestingly, these relationships were not observed in patients with low LAT1 expression. This discrepancy may reflect the fact that LAT1 has been shown to be estrogen-dependent in endocrine-responsive cells [51, 52]. Therefore, it is likely that more of the tumors in the low LAT1 group were triple-negative breast cancers, which generally have a poor prognosis independently of menopausal status. We recognize that worse survival is expected over the more than 10-year duration of follow-up in the datasets analyzed in postmenopausal patients, who are older and at greater risk for numerous conditions than their younger counterparts. Thus, the fact that survival differences were not observed in the LAT1 group implies that a regulator of LAT1 expression—such as estrogen—may obscure expected differences in survival. Additionally, we observed a positive correlation (r < 0.5) between BMI and basophil, eosinophil, monocyte, and lymphocyte counts in postmenopausal patients, a finding not seen in premenopausal patients. On the contrary, the opposite was observed in premenopausal patients. Premenopausal patients experience heightened 17β-estradiol levels, dampening obesity-induced inflammation, whereas postmenopausal patients (and those with obesity) have higher levels of estrone, stimulating inflammation. The imbalance between estrone and 17β-estradiol levels that occurs after menopause results in the release of cytokines and the recruitment of nearby immune cells, which likely explains this correlation only being observed in postmenopausal patients [53]. A limitation of our survival data is that LAT1 expression was measured, but the patients' levels of inflammatory cytokines were not. Because of this, we were unable to observe the association between LAT1 expression and cytokine circulation, but future studies should pursue this.

Past literature has shown LAT1 to be involved in endocrine therapy resistance in ER+ breast cancer patients [51, 54, 55]. In addition, LAT1 expression in HER2- patients has been shown to contribute to treatment resistance [56]. Because of the significance of these two molecular subtypes, we sought to observe if menopausal status and LAT1 expression impacted patient survival. We did not observe significant differences in survival in any of the breast cancer subtypes between low and high LAT1 expression; however, low LAT1 expression strongly tended to be a favorable prognostic factor in postmenopausal patients with ER+ (p = 0.05) and HER2- tumors (p = 0.10) In contrast, LAT1 did not approach significance as a prognostic factor in premenopausal patients with ER+ (p = 0.34) or HER2- tumors (p = 0.65). These data highlight the importance of considering molecular subtype and menopausal status when examining LAT1 as a prognostic factor in breast cancer. These are considerations future efforts should make as there exists minimal work outside the current study on how both molecular subtype and menopausal status stratify how LAT1 expression affects breast cancer survival. If so, interventions targeting specific LAT1—such as JPH203, a LAT1 inhibitor that has recently shown to be tolerated in patients with advanced solid tumors [57]—could be administered using precision medicine approaches in patients with breast cancer and potentially other tumors.

[18]F-fluorodeoxyglucose ([18]F-FDG) has been the traditional radiotracer utilized in cancer research, and it is still used in the majority of tumor radiotracer analyses. Indeed, the Positron Emission Tomography Response Criteria in Solid Tumors (PERCIST) is based on the use of [18]F-FDG as the radiotracer [56, 58]. However, although high [18]F-FDG uptake correlates with poor prognosis in numerous tumor types, including breast cancer [59–64], it is not a direct readout of tumor proliferative activity. For this, it is necessary to utilize a tracer such as [18]F-FLT, an analog of thymidine which is phosphorylated by thymidine kinase prior to incorporation into DNA during cell replication. Because our study employs [18]F-FLT imaging as a more direct readout of tumor proliferation rather than [18]F-FDG, we provide an analysis that

has previously been insufficiently explored. We observe differences in the strength of the correlation between Ki-67 and $^{18}$F-FLT uptake in pre- and postmenopausal patients: in postmenopausal patients, Ki-67 significantly positively correlated with $^{18}$F-FLT SUV$_{Mean}$, SUV$_{Peak}$, and SUV$_{Max}$, whereas in premenopausal patients, Ki-67 insignificantly positively correlated with $^{18}$F-FLT. These data are consistent with prior studies in which the correlation between Ki-67 and $^{18}$F-FLT was found to be relatively weak and dependent on clinical variables (pre- or post-treatment timing, hormone receptor status) [62, 63]. Surprisingly, BMI barely correlated with either Ki-67 or $^{18}$F-FLT, which may indicate that obesity is more involved in the appearance—and potentially recurrence—of cancer rather than its progression once a tumor is already established. Further work will be required to better understand the nuanced relationships between these clinical variables. Additionally, it will be important to understand the relationship between LAT1 expression, $^{18}$F-FLT uptake, and clinical variables including BMI and—better yet [64]—adiposity, as well as additional molecular factors that were not available in the datasets analyzed. In fact, to our knowledge, there are no studies correlating LAT1 expression to all 3 types of $^{18}$F-FLT SUVs. We recognize that a limitation of our study is that BMI is not the best metric for obesity. In the datasets analyzed, there were no clinical data including possible alternatives for BMI like visceral adiposity, so we did not have an alternative to relying on BMI. This limitation exists largely because breast cancer imaging occurs at levels that typically do not allow calculation of visceral adipose tissue mass. Correlating $^{18}$F-FLT uptake to both gene expression and a broad range of anthropometric indices, including visceral adiposity, will be of great interest in future studies. Also, $^{18}$F-FLT uptake and Ki-67 values should be correlated to LAT1 expression to explore its role in tumor proliferation. This would make sense considering LAT1 expression has already been established in the activation of the mTOR pathway, promoting cell proliferation in breast cancer [43]. Finally, future clinical trials will be required to establish the utility of LAT1 as a biomarker for breast cancer prognosis, particularly in association with other clinical factors: survival data in the datasets analyzed were limited, but will be important to examine in forthcoming studies.

## Conclusion

Through our analyses, we show that although the extent to which this occurs is stratified by menopausal status, LAT1 expression worsens breast cancer prognosis, bolstering the role of amino acid metabolism in tumor energetics, an aspect of the literature that has been underexplored. Using various clinical variables, we correlated tumor proliferation, body composition, and immune cell populations to identify the complex relationships underlying metabolism, immune surveillance, and cancer progression. Future studies should aim to utilize a wider variety of immune cell types and metrics for body composition, while further segmenting patients based on breast cancer subtype and menopausal status, to gain a more comprehensive understanding of the findings we establish here. In addition, we speculate that future studies should target LAT1 or its heavy chain partner 4F2hc to inhibit the LAT1-4F2hc complex, interventions that may plausibly improve patient outcomes.

## Supporting information

**S1 Fig. Prognosis in TCGA BRCA patients with ER+ tumors.** Patients were separated into (A) low (< 10.09 FPKM) and (B) high (> = 10.09 FPKM) LAT1 expression groups. Survival is observed for each menopausal status: premenopausal, postmenopausal, and peri-menopausal. (TIF)

**S2 Fig. Prognosis in TCGA BRCA patients with HER2- tumors.** Patients were separated into (A) low ($<$ 10.38 FPKM) and (B) high ($>$ = 10.38 FPKM) expression groups. Survival is observed for each menopausal status: premenopausal, postmenopausal, and peri-menopausal. (TIF)

**S1 Table. Mann-Whitney U tests identify no significant differences between the log2 fold change in Ki-67 and [18]F-FLT uptake in pre- or postmenopausal patients.** P-values are shown.
(DOCX)

**S2 Table. Mann-Whitney U tests identify differences between the log2 fold change in BMI and SUV$_{Mean}$ in postmenopausal patients, but no other proliferative markers differed in pre- or postmenopausal patients.**
(DOCX)

**S3 Table. Mann-Whitney U and student's t-tests identify differences between immune cells and proliferative markers in premenopausal patients.** The log2 fold change of each parameter was compared. The comparisons' p-values are shown above. Shapiro-Wilk tests were used to assess the data's normality. Based on these results, normally distributed data were compared using the Student's t-test, and all other analyses used the Mann-Whitney test. [a] These comparisons use the Student's t-test.
(DOCX)

**S4 Table. Mann-Whitney U tests identify differences between immune cells and proliferative markers in postmenopausal patients.** The log2 fold change of each parameter was compared. The comparisons' p-values are shown above. Significant results are bolded, and marginally significant results are bolded and italicized.
(DOCX)

## Acknowledgments

The authors thank Dr. Gang Peng for helpful discussions lending his biostatistical expertise to the analysis of data in this manuscript.

## Author Contributions

**Conceptualization:** Gautham Ramshankar, Rachel J. Perry.

**Data curation:** Gautham Ramshankar.

**Formal analysis:** Gautham Ramshankar.

**Funding acquisition:** Rachel J. Perry.

**Investigation:** Gautham Ramshankar, Ryan Liu.

**Supervision:** Rachel J. Perry.

**Writing – original draft:** Gautham Ramshankar.

**Writing – review & editing:** Rachel J. Perry.

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
