## [Decision Letter · Decision Letter 0]

18 Aug 2023

PONE-D-23-21115The Influence of the Amino Acid Transporter LAT1 on Patient Prognosis and the Relationships between Tumor Immunometabolic and Proliferative Features Depend on Menopausal Status in Breast CancerPLOS ONE

Dear Dr. Perry,

Thank you for submitting your manuscript to PLOS ONE. After careful consideration, we feel that it has merit but does not fully meet PLOS ONE’s publication criteria as it currently stands. Therefore, we invite you to submit a revised version of the manuscript that addresses the points raised during the review process. 

As you can see that the reviewers have raised some serious concerns. I invite you to resubmit the manuscript after addressing all the concerns.

We look forward to receiving your revised manuscript.

Kind regards,

Pankaj K Singh, Ph.D.

Academic Editor

PLOS ONE

“The authors are grateful for awards from the Lion Heart Foundation and from the Yale Cancer Center, which supported this research.”

“The authors are grateful for awards from the Lion Heart Foundation and from the Yale Cancer Center (both to R.J.P.), which supported this research. The funders had no role in study design, data collection and analysis, decision to publish, or preparation of the manuscript.”

4. Please upload a new copy of Figures 1, 2, 3, 4 and 5 as the details are not clear. Please follow the link for more information: https://blogs.plos.org/plos/2019/06/looking-good-tips-for-creating-your-plos-figures-graphics/" https://blogs.plos.org/plos/2019/06/looking-good-tips-for-creating-your-plos-figures-graphics/

Reviewers' comments:

Reviewer's Responses to Questions

**Comments to the Author**

1. Is the manuscript technically sound, and do the data support the conclusions?

Reviewer #1: Yes

Reviewer #2: No

2. Has the statistical analysis been performed appropriately and rigorously? 

Reviewer #1: Yes

Reviewer #2: No

3. Have the authors made all data underlying the findings in their manuscript fully available?

Reviewer #1: Yes

Reviewer #2: Yes

4. Is the manuscript presented in an intelligible fashion and written in standard English?

Reviewer #1: Yes

Reviewer #2: Yes

5. Review Comments to the Author

Reviewer #1: In this article, using transcriptomic data analysis, the authors demonstrate that high tumor LAT1 expression predicts abbreviated survival in postmenopausal breast cancer patients compared to peri- or premenopausal patients. The authors also discuss about tumor immunometabolism and proliferation and show that these features are associated with pre and postmenopausal statuses of breast cancer patients. They show that tumor Ki-67 staining significantly and positively correlated with 18F-FLT uptake in postmenopausal patients.

LAT1 expression is already well established as an independent poor prognostic factor in invasive breast cancer and other cancer types. While there are many shortcomings in the article which makes conclusions rather confounding, like the use of BMI as an obesity marker and including weak insignificant correlations of immune cells with SUV, and Ki-67, the correlation of LAT1 expression with the menopausal status is clinically relevant and significant. However, the lack of inclusion of other molecular factors in this study raises concerns, see below, and undermines the clinical association and utility of this work.

1. The authors do not show if LAT1 expression correlates with menopausal status in breast cancer subtypes especially in ER+, and HER2- tumors. There is a huge body of literature suggesting LAT1 as an estrogen responsive gene and that LAT1 is responsible for endocrine therapy resistance. Likewise, HER2- tumors have been shown to have high expression of LAT1 and do not respond to current therapies. It would be interesting to see if LAT1 expression can stratify pre and post-menopausal breast cancer patient survival in these subtypes. If so, these cohorts can benefit from drugs like JPH203, currently in clinical trial, targeting LAT1 activity.

2. Postmenopausal patients have increased systemic circulation of inflammatory cytokines such as IL-6, IL-1beta, and TNFa. Coincidently, these cytokines are also known to regulate the expression of LAT1 and can promote tumor-immune infiltration. The authors do not show if any of these post-menopausal cytokines correlate with LAT1 expression in the same data sets as a casual mechanistic link for LAT1 dependent abbreviated survival? If so, this rationally explains the significant worse survival association of LAT1 in postmenopausal patients.

3. Both Ki-67 and 81F-FLT uptake are good quantitative markers of tumor proliferation. The authors show a strong association of Ki-67 with 18F-FLT uptake in postmenopausal patients. Does Ki-67 and 18F-FLT uptake show a significantly positive association with LAT1 expression in postmenopausal patients in the same dataset? This is because LAT1 is known to activate mTORC1 through uptake of leucine and mTOR activation drives cell proliferation.

4. In the title, the authors declare that the relationships between tumor immunometabolic and proliferative features depend on the menopausal status in breast cancer. While the metabolic and proliferative features do significantly correlate with menopausal status, the immune cells demonstrate only weak correlation and is not significant. The weak positive correlation in postmenopausal patients could again be because of the high levels of systemic cytokines in these cohort.

The authors need to address these concerns to strengthen the use of LAT1 as a clinical biomarker for breast cancer patient prognosis.

Reviewer #2: The manuscript by Ramshankar et al. deals with the relationship among various clinical and genetic factors including fluorothymidine uptake, immune cell infiltration, body weight, LAT1 expression, and cell proliferation markers (e.g. ki67). This manuscript is unclear in term of its scientific significance and message. All data are not connected as one story. Thus it’s very hard to understand the authors’ opinion. In addition, some findings are not novel. For example, the correlation between LAT1 expression and poor prognosis has been reported in breast cancer patients (PMID: 35177712, PMID: 29566741). Most of all, the main reason why I decide to reject this manuscript is that the study was not scientifically sound, and the data interpretation was flawed.

• In Figure 1, the authors used the Pearson correlation to analyze the association among various clinical factors. The Pearson correlation is for measuring the linear relationship between two variables. Correlation can be biased. In my opinion, multivariate approach is more suitable in clinical data. The author should consider multicollinearity.

• Regarding to the data analysis in Figure 1, the author stated “basophil, eosinophil, neutrophil, monocyte, and lymphocyte counts insignificantly negatively correlated” in line 262, “White blood cell counts insignificantly positively correlated” in line 263. I can’t understand this statement. White blood cell is consisted of five different subtypes of cells – basophil, eosinophil, neutrophil, monocyte, and lymphocyte. I am deeply concerned that the author may have provided ambiguous information to the academic community or patients based on data that was interpreted with unclear statistical methods with limited medical knowledge. This kind of analysis was repeated in Line 267 – 277.

• In Figure 2 and 3, the author monitored the survival rate up to 8,605 days. It’s almost 23 years after the initial treatment. I am not an expert in survival monitoring, but wondering it’s too long to interpret the event is the disease-related.

• It is not clear that the authors included and excluded the appropriate patients for this study.

• The introduction contains unclear or inaccurate information. For example, “Nearly 30% of breast cancer deaths are caused by modifiable risk factors like excess body weight and alcohol consumption [2]” in Line 37. Please check the reference [2] and the original reference again.

Overall, I do not think this manuscript is suitable for publication.

6. PLOS authors have the option to publish the peer review history of their article (what does this mean?). If published, this will include your full peer review and any attached files.

Reviewer #1: **Yes: **Surajit Sinha

Reviewer #2: No

---

## [Author Response · Author response to Decision Letter 0]

31 Aug 2023

We thank the editors and reviewers for their comments, which have prompted us to make edits that, in our view, substantially improve this manuscript. All comments are reproduced verbatim in bold in our response to reviewers, with our responses in unbolded text, but unfortunately this does not come through on the website.

Thank you for these links. We have followed these templates to ensure that our manuscript meets the journal’s style requirements.

We have edited the Methods to include the following statement: 

“Because only publicly available, deidentified data were analyzed, and participants, all of whom were adults, gave written consent for their data to be used, deidentified, in public repositories, separate ethical approval is not required for these or other datasets analyzed in this manuscript. Data sharing via TCIA is approved under the supervision of the University of Arkansas for Medical Sciences (UAMS) Institutional Review Board (IRB # 205568), and informed consent was provided by the patients for their data to be shared with TCIA; however, the details of the consent process are not available to us. Because only publicly available, deidentified data were analyzed, and we had no information about the patients whose data were analyzed, separate ethical approval was not sought. All data are submitted to the TGCA in accordance with the submitter’s institutional policies, including IRB approval and informed consent provided by the patients for their data to be submitted, deidentified, to the TCGA. However, the details of the consent process are not available to us. We did not seek separate IRB approval because our use of these deidentified data are covered under these IRB approvals.”

This text has been included in the Ethics Statement field of the submission form.

Thank you for providing this helpful information.

“The authors are grateful for awards from the Lion Heart Foundation and from the Yale Cancer Center, which supported this research.”

“The authors are grateful for awards from the Lion Heart Foundation and from the Yale Cancer Center (both to R.J.P.), which supported this research. The funders had no role in study design, data collection and analysis, decision to publish, or preparation of the manuscript.”

We have removed the funding-related text from the acknowledgments in the manuscript, and have included the funding statement (which we do not intend to change) in our cover letter. The tool to build the funding statement in the online submission system does not provide a place to comment that the funders had no role in study design, data collection and analysis, decision to publish, or preparation of the manuscript, so we have removed this.

4. Please upload a new copy of Figures 1, 2, 3, 4 and 5 as the details are not clear. Please follow the link for more information: https://blogs.plos.org/plos/2019/06/looking-good-tips-for-creating-your-plos-figures-graphics/" https://blogs.plos.org/plos/2019/06/looking-good-tips-for-creating-your-plos-figures-graphics/

We have uploaded new copies of each of these figures at substantially higher resolution. 

 

Reviewer Comments:

Reviewer #1: In this article, using transcriptomic data analysis, the authors demonstrate that high tumor LAT1 expression predicts abbreviated survival in postmenopausal breast cancer patients compared to peri- or premenopausal patients. The authors also discuss about tumor immunometabolism and proliferation and show that these features are associated with pre and postmenopausal statuses of breast cancer patients. They show that tumor Ki-67 staining significantly and positively correlated with 18F-FLT uptake in postmenopausal patients.

LAT1 expression is already well established as an independent poor prognostic factor in invasive breast cancer and other cancer types. While there are many shortcomings in the article which makes conclusions rather confounding, like the use of BMI as an obesity marker and including weak insignificant correlations of immune cells with SUV, and Ki-67, the correlation of LAT1 expression with the menopausal status is clinically relevant and significant. However, the lack of inclusion of other molecular factors in this study raises concerns, see below, and undermines the clinical association and utility of this work.

We thank the reviewer for their time spent examining this manuscript and for their helpful comments. Although discussed in more detail in response to the specific comments below, we agree with the reviewer’s important point that BMI as an obesity marker has shortcomings, as we had commented in the manuscript:

“We recognize that a limitation of our study is that BMI is not the best metric for obesity. In the datasets analyzed, there were no clinical data including possible alternatives for BMI like visceral adiposity, so we did not have an alternative to relying on BMI[…] Correlating 18F-FLT uptake to both gene expression and a broad range of anthropometric indices, including visceral adiposity, will be of great interest in future studies.”

To clarify the reason we did not include these analyses in the current study - an important point highlighted by the reviewer - we have added an additional comment within the statement above, clarifying that without imaging studies that allow quantification of visceral adiposity, BMI is the best parameter we can analyze with data available from patients with breast cancer:

“This limitation exists largely because breast cancer imaging occurs at levels that typically do not allow calculation of visceral adipose tissue mass.”

With regard to the inclusion of other molecular factors, we certainly agree that this is a limitation of the study that results from the fact that we are restricted to data available in the extant datasets, and we have included a comment (following “as well as”) about this in the revised manuscript:

“Additionally, it will be important to understand the relationship between LAT1 expression, 18F-FLT uptake, and clinical variables including BMI and - better yet - adiposity, as well as additional molecular factors that were not available in the datasets analyzed.”

1. The authors do not show if LAT1 expression correlates with menopausal status in breast cancer subtypes especially in ER+, and HER2- tumors. There is a huge body of literature suggesting LAT1 as an estrogen responsive gene and that LAT1 is responsible for endocrine therapy resistance. Likewise, HER2- tumors have been shown to have high expression of LAT1 and do not respond to current therapies. It would be interesting to see if LAT1 expression can stratify pre and post-menopausal breast cancer patient survival in these subtypes. If so, these cohorts can benefit from drugs like JPH203, currently in clinical trial, targeting LAT1 activity.

Thank you for this helpful suggestion. Using the TCGA BRCA dataset, we have assessed the impact of LAT1 expression on prognosis in premenopausal and postmenopausal patients with ER+ and HER2- tumors as 4 new Kaplan-Meier plots in Supplementary Figures 1 and 2:

Supplementary Figure 1

Supplementary Figure 2

In addition, we analyzed the aforementioned data separately to compare survival in patients with low and high LAT1 expression specifically in ER+ and HER2- negative tumors; these data, now shown in the new Figures 6 and 7, demonstrate that while there was no statistically significant difference in survival, high LAT1 expression much more strongly tended to predict poor prognosis in postmenopausal women with ER+ breast cancer (new Fig 6)

Figure 6

as well as in postmenopausal women with HER2- breast cancer (new Fig 7):

Figure 7

Taken together these data suggest that menopausal status affects the impact of LAT1 on survival in patients with both ER+ and HER2- tumors, though prospective studies will be required to confirm or refute this. TCGA BRCA is the only dataset from those we used that includes patients’ menopausal statuses, thus limiting the data available for these analyses. It is possible that some of the strong tendencies without reaching statistical significance in the aforementioned analyses may be affected by the relatively low sample numbers in these analyses.

2. Postmenopausal patients have increased systemic circulation of inflammatory cytokines such as IL-6, IL-1beta, and TNFa. Coincidently, these cytokines are also known to regulate the expression of LAT1 and can promote tumor-immune infiltration. The authors do not show if any of these post-menopausal cytokines correlate with LAT1 expression in the same data sets as a casual mechanistic link for LAT1 dependent abbreviated survival? If so, this rationally explains the significant worse survival association of LAT1 in postmenopausal patients.

The survival datasets we utilized on Xena Functional Genomics Browser did not measure patients’ inflammatory cytokines. We agree that this would have been a valuable correlation to explore so we have added a comment addressing this idea to the Discussion section, which can urge future studies to analyze this:

“A limitation of our survival data is that LAT1 expression was measured, but the patients’ levels of inflammatory cytokines were not. Because of this, we were unable to observe the association between LAT1 expression and cytokine circulation but future studies should pursue this.”

3. Both Ki-67 and 81F-FLT uptake are good quantitative markers of tumor proliferation. The authors show a strong association of Ki-67 with 18F-FLT uptake in postmenopausal patients. Does Ki-67 and 18F-FLT uptake show a significantly positive association with LAT1 expression in postmenopausal patients in the same dataset? This is because LAT1 is known to activate mTORC1 through uptake of leucine and mTOR activation drives cell proliferation.

The “ACRIN-FLT-Breast (ACRIN 6688)” dataset which offered patients’ Ki-67 values and tumor 18F-FLT uptake did not include LAT1 expression data. Likewise, the survival datasets which offered LAT1 expression data did not include patients’ Ki-67 values and tumor 18F-FLT uptake. Because of this, we were unable to correlate tumor proliferative factors to LAT1 expression. However, we have added a comment in the manuscript to highlight that this would be a worthwhile correlation to perform in the future:

Also, 18F-FLT uptake and Ki-67 values should be correlated to LAT1 expression to explore its role in tumor proliferation. This would make sense considering LAT1 expression has already been established in the activation of the mTOR pathway, promoting cell proliferation in breast cancer [43].

4. In the title, the authors declare that the relationships between tumor immunometabolic and proliferative features depend on the menopausal status in breast cancer. While the metabolic and proliferative features do significantly correlate with menopausal status, the immune cells demonstrate only weak correlation and is not significant. The weak positive correlation in postmenopausal patients could again be because of the high levels of systemic cytokines in these cohort.

We agree with the reviewer’s insightful point, and therefore have changed the title to more accurately reflect the findings of this analysis: “The Association between the Amino Acid Transporter LAT1, Tumor Immunometabolic and Proliferative Features and Menopausal Status in Breast Cancer.” Additionally, we have added a comment to the Discussion regarding the need for future studies to correlate clinical variables and, specifically, inflammatory cytokines to outcomes: 

“Additionally, future clinical trials will be required to establish the utility of LAT1 as a biomarker for breast cancer prognosis, particularly in association with other clinical factors: survival data in the datasets analyzed were limited, but will be of great interest in forthcoming studies.”

We hope that these revisions will help to position this manuscript as scientifically sound and well communicated, in keeping with the criteria for publication in PLOS One.

The authors need to address these concerns to strengthen the use of LAT1 as a clinical biomarker for breast cancer patient prognosis.

Unfortunately, as mentioned in our point-by-point responses to each comment, we are limited by the available data and are not able to perform all of the analyses recommended because of the lack of available data required to establish the use of LAT1 as a clinical biomarker for breast cancer prognosis. Indeed, this would require a clinical trial! Instead, we intend for this work to clarify the association between LAT1 and other prognostic and clinical factors in breast cancer, and have added a comment to the discussion to address this point as well as the point immediately above from Reviewer 1. We respectfully hope that these revisions will, in the reviewer’s view, be sufficient to justify the manuscript as appropriate for publication according to PLOS One’s criteria: “Experiments, statistics, and other analyses are performed to a high technical standard and are described in sufficient detail,” and “Conclusions are presented in an appropriate fashion and are supported by the data,” as well as that the work be original and the data presentation be clear. 

 

Reviewer #2: The manuscript by Ramshankar et al. deals with the relationship among various clinical and genetic factors including fluorothymidine uptake, immune cell infiltration, body weight, LAT1 expression, and cell proliferation markers (e.g. ki67). This manuscript is unclear in term of its scientific significance and message. All data are not connected as one story. Thus it’s very hard to understand the authors’ opinion. In addition, some findings are not novel. For example, the correlation between LAT1 expression and poor prognosis has been reported in breast cancer patients (PMID: 35177712, PMID: 29566741). Most of all, the main reason why I decide to reject this manuscript is that the study was not scientifically sound, and the data interpretation was flawed.

We sincerely thank the reviewer for their time spent on this manuscript, and for their comments which have helped us to improve the manuscript for resubmission as invited by the editor. In keeping with PLOS One’s criteria, we aim to improve the manuscript to ensure that it is scientifically sound and well-presented. 

• In Figure 1, the authors used the Pearson correlation to analyze the association among various clinical factors. The Pearson correlation is for measuring the linear relationship between two variables. Correlation can be biased. In my opinion, multivariate approach is more suitable in clinical data. The author should consider multicollinearity.

We used the Pearson correlation because we hoped to observe the linear relationship between each pair of clinical variables, as Reviewer 2 described. The reviewer’s concern that clinical data may have multicollinearity is certainly reasonable, so we consulted Dr. Gang Peng, formerly a Yale Cancer Center biostatistician (now faculty at Indiana University), who agreed with the use of the Pearson correlation test in this context. To quote his email: 

“If you want to show the association between two variables (x, y), correlation test is enough (Pearson for linear and Spearman for nonlinear).”

In the interest of full transparency, the complete email exchange between Dr. Peng and the corresponding author is shown at the end of this response to reviewer comments on the following page of this rebuttal.

• Regarding to the data analysis in Figure 1, the author stated “basophil, eosinophil, neutrophil, monocyte, and lymphocyte counts insignificantly negatively correlated” in line 262, “White blood cell counts insignificantly positively correlated” in line 263. I can’t understand this statement. White blood cell is consisted of five different subtypes of cells – basophil, eosinophil, neutrophil, monocyte, and lymphocyte. I am deeply concerned that the author may have provided ambiguous information to the academic community or patients based on data that was interpreted with unclear statistical methods with limited medical knowledge. This kind of analysis was repeated in Line 267 – 277.

In addition to basophil, eosinophil, neutrophil, monocyte, and lymphocyte counts, the clinical data we used offered white blood cell counts as a separate measured value from the basophil, eosinophil, neutrophil, monocyte, and lymphocyte counts, but we have removed WBC counts for clarity:

• In Figure 2 and 3, the author monitored the survival rate up to 8,605 days. It’s almost 23 years after the initial treatment. I am not an expert in survival monitoring, but wondering it’s too long to interpret the event is the disease-related.

In these figures, we monitored the survival rate up to 8605 days because that is where patient survival data ends in the datasets used. 10-year survival in breast cancer patients is 80%: this is a good problem to have, and means that long-term tracking of outcomes is important. However, as the reviewer highlights we cannot be sure that mortality at this late timepoint is cancer-related, and have added a comment addressing this important point:

“We recognize that the long followup period prevents us from concluding with certainty that mortality is breast cancer-related, but even if mortality were unrelated to cancer at this time point, the utility of LAT1 as a prognostic factor remains important.”

• It is not clear that the authors included and excluded the appropriate patients for this study.

The inclusion criteria for the 18F-FLT PET-CT image analysis, as mentioned in the Methods section, are the following:

“We analyzed the scans of all patients with a menopausal status, height, weight, and 5 clear CT slices (i.e., slices in which the primary breast tumor could be identified and its corresponding SUV values could be generated) present in the dataset. 58 of the 90 enrolled patients in the ACRIN clinical trial met these criteria, and all were analyzed.”

No patients whose data included menopausal status, height, weight, and 5 clear CT slices were excluded. We are unsure as to Reviewer 2’s specific concern and would be happy to provide more information to clarify as needed.

• The introduction contains unclear or inaccurate information. For example, “Nearly 30% of breast cancer deaths are caused by modifiable risk factors like excess body weight and alcohol consumption [2]” in Line 37. Please check the reference [2] and the original reference again.

We appreciate the reviewer for identifying this mistake. We have corrected this: “Nearly 30% of breast cancer cases are caused by modifiable risk factors like excess body weight and alcohol consumption [2].”

Overall, I do not think this manuscript is suitable for publication.

We thank the reviewer for his/her comments, which have catalyzed edits that substantially improve the manuscript. We hope the reviewer will believe that the manuscript is suitable for publication after these revisions in keeping with PLOS One’s editorial criteria.

---

## [Decision Letter · Decision Letter 1]

25 Sep 2023

PONE-D-23-21115R1The Association between the Amino Acid Transporter LAT1, Tumor Immunometabolic and Proliferative Features and Menopausal Status in Breast CancerPLOS ONE

Dear Dr. Perry,

Thank you for submitting your manuscript to PLOS ONE. After careful consideration, we feel that it has merit but does not fully meet PLOS ONE’s publication criteria as it currently stands. Therefore, we invite you to submit a revised version of the manuscript that addresses the additional two points raised by Reviewer 1 during the review process.

We look forward to receiving your revised manuscript.

Kind regards,

Pankaj K Singh, Ph.D.

Academic Editor

PLOS ONE

Journal Requirements:

Reviewers' comments:

Reviewer's Responses to Questions

**Comments to the Author**

1. If the authors have adequately addressed your comments raised in a previous round of review and you feel that this manuscript is now acceptable for publication, you may indicate that here to bypass the “Comments to the Author” section, enter your conflict of interest statement in the “Confidential to Editor” section, and submit your "Accept" recommendation.

Reviewer #1: All comments have been addressed

Reviewer #2: (No Response)

2. Is the manuscript technically sound, and do the data support the conclusions?

Reviewer #1: Yes

Reviewer #2: Yes

3. Has the statistical analysis been performed appropriately and rigorously? 

Reviewer #1: Yes

Reviewer #2: N/A

4. Have the authors made all data underlying the findings in their manuscript fully available?

Reviewer #1: Yes

Reviewer #2: Yes

5. Is the manuscript presented in an intelligible fashion and written in standard English?

Reviewer #1: Yes

Reviewer #2: Yes

6. Review Comments to the Author

Reviewer #1: (No Response)

Reviewer #2: The authors have addressed the concerns arisen from data analysis and interpretation. First, statistical analysis in this manuscript were revised by the authors, remaining a few concerns that all studies have. The authors explained properly on the analysis. Second, the authors revised the ambiguous medical interpretations by describing clearly and adding the limitations and shortcomings about their result and interpretation. This study suggests that LAT1 expression status can be a possible biomarker in breast cancer patients. They also suggested a few clinical factors that highly related to breast cancer incidence such as a proliferation status, obesity, immune / inflammation status. Despite the limitation of available clinical data representing the risk factors, the authors demonstrated the best way predicting the prognosis of breast cancer patients with factors including LAT1 expression. Since the authors mentioned the necessity of future studies to validate this finings, the readers can interpret and assume the intention and purpose of this article. The article also suggested the correlation among the factors before / after menopause, indicating this article has a novelty in this field. In addition, this article is of importance to describe the method and result of early biomarker study in breast cancer research. I recommend to publish this article with the minor revision.

1. The purpose of Figure 1 is to show the association between two factors. Is it necessary to show how much different between two factors in Supple Table 5-8? Also, the authors performed the U-test (non-parametric) instead of T-test (parametric), however in Figure 1, the authors performed the pearson test (parametric) instead of spearman (non-parametric) with the same data. In my suggestion, Table 5-8 is not necessary.

2. I think it's better to describe more details in the legend for figure 8 . Please add conclusions or explains with the fig 8 legend title for the readers who read the summary figure first.

7. PLOS authors have the option to publish the peer review history of their article (what does this mean?). If published, this will include your full peer review and any attached files.

Reviewer #1: **Yes: **Surajit Sinha

Reviewer #2: No

---

## [Author Response · Author response to Decision Letter 1]

26 Sep 2023

We thank the reviewers and editors for their time spent evaluating this manuscript. We are delighted that the reviewers and editor find the manuscript of interest, and that Reviewer 1 feels that we have addressed all of their comments. We have edited the manuscript in response to Reviewer 2’s remaining comments as described below.

Reviewer #2: The authors have addressed the concerns arisen from data analysis and interpretation. First, statistical analysis in this manuscript were revised by the authors, remaining a few concerns that all studies have. The authors explained properly on the analysis. Second, the authors revised the ambiguous medical interpretations by describing clearly and adding the limitations and shortcomings about their result and interpretation. This study suggests that LAT1 expression status can be a possible biomarker in breast cancer patients. They also suggested a few clinical factors that highly related to breast cancer incidence such as a proliferation status, obesity, immune / inflammation status. Despite the limitation of available clinical data representing the risk factors, the authors demonstrated the best way predicting the prognosis of breast cancer patients with factors including LAT1 expression. Since the authors mentioned the necessity of future studies to validate this finings, the readers can interpret and assume the intention and purpose of this article. The article also suggested the correlation among the factors before / after menopause, indicating this article has a novelty in this field. In addition, this article is of importance to describe the method and result of early biomarker study in breast cancer research. I recommend to publish this article with the minor revision.

We greatly appreciate the reviewer’s many positive comments on our revision, indicating that we have addressed most of the issues they raised. We are delighted that the reviewer believes “this article is of importance to describe the method and result of early biomarker study in breast cancer research.” We have revised according to each of the reviewer’s comments as described below, and after these revisions hope the article will be deemed acceptable for publication.

1. The purpose of Figure 1 is to show the association between two factors. Is it necessary to show how much different between two factors in Supple Table 5-8? Also, the authors performed the U-test (non-parametric) instead of T-test (parametric), however in Figure 1, the authors performed the pearson test (parametric) instead of spearman (non-parametric) with the same data. In my suggestion, Table 5-8 is not necessary.

We thank the reviewer for these comments, and in accordance, have removed Supplemental Tables 5-8. We agree that considering the presentation of data in Figure 1, Supplemental Tables 5-8 are not necessary.

2. I think it's better to describe more details in the legend for figure 8 . Please add conclusions or explains with the fig 8 legend title for the readers who read the summary figure first.

We have added additional description as follows: “The current study analyzed 18F-FLT uptake, immune infiltrate levels, body weights, LAT1 expression, and tumor proliferative factors observed in breast cancer patients.”

---

## [Editor Report · Decision Letter 2]

26 Sep 2023

The Association between the Amino Acid Transporter LAT1, Tumor Immunometabolic and Proliferative Features and Menopausal Status in Breast Cancer

PONE-D-23-21115R2

Dear Dr. Perry,

We’re pleased to inform you that your manuscript has been judged scientifically suitable for publication and will be formally accepted for publication once it meets all outstanding technical requirements.

Kind regards,

Pankaj K Singh, Ph.D.

Academic Editor

PLOS ONE
---

## [Editor Report · Acceptance letter]

3 Oct 2023

PONE-D-23-21115R2 

The Association between the Amino Acid Transporter LAT1, Tumor Immunometabolic and Proliferative Features and Menopausal Status in Breast Cancer 

Dear Dr. Perry:

I'm pleased to inform you that your manuscript has been deemed suitable for publication in PLOS ONE. Congratulations! Your manuscript is now with our production department. 

Kind regards, 

on behalf of

Dr. Pankaj K Singh 

Academic Editor

PLOS ONE